# OpenReview forum: "RWKU: Benchmarking Real-World Knowledge Unlearning for Large Language Models"
_NeurIPS.cc/2024/Datasets_and_Benchmarks_Track — NeurIPS 2024 Track Datasets and Benchmarks Poster_

### Official Review · Reviewer_nhq4 · 2024-07-21
**review on paper 93**

**Rating:** 7
**Confidence:** 4
**Correctness:** yes
**Clarity:** yes

**Review:**

See below.

**Strengths:**

The authors present a new benchmark test, RWKU, designed and evaluated comprehensively for the unlearning  ability of LLMs, including the task setup, knowledge sources, and evaluation framework.
The authors designed forgetting sets and retention sets to evaluate the model's capabilities in a variety of real-world applications. The forgetting set tested forgetting effects through four membership inference attack (MIA) methods and nine adversarial attack probes. The retention set assessed localization and utility, including neighborhood perturbation, general competence, reasoning ability, truthfulness, factuality, and fluency.
The paper writing was good and relatively solid.

**Additional Feedback:**

See above

**Documentation:**

yes

**Ethics:**

No need

**Limitations:**

yes

**Opportunities For Improvement:**

The authors chose 200 celebrities as forgetting targets, suggesting that forgetting targets could be more diverse and not limited to celebrities.
Two pieces of work related to knowledge forgetting/unlearning were omitted:
[1] Liu, Zheyuan, et al. "Towards safer large language models through machine unlearning." arXiv preprint arXiv:2402.10058 (2024).
[2] Ni, Shiwen, et al. "Forgetting before learning: Utilizing parametric arithmetic for knowledge updating in large language models." arXiv preprint arXiv:2311.08011 (2023).

**Relation To Prior Work:**

yes

**Summary And Contributions:**

This paper focuses on proposing a real-world knowledge unlearning benchmark (RWKU) for large language models (LLMs), which aims to address the problem that LLMs inevitably memorize sensitive, copyrighted, and harmful knowledge during training. An effective forgetting method is proposed and its effectiveness is verified through a series of experiments.

---

> ### Author Rebuttal · Authors · 2024-08-15
>
> Thanks for your careful and insightful reviews. Your professional reviews offer us great advice towards writing a more comprehensive and competitive paper!
>
>
>
> > The authors chose 200 celebrities as forgetting targets, suggesting that forgetting targets could be more diverse and not limited to celebrities.
>
>
> We completely agree with your suggestion that unlearning targets could be more diverse and not limited to celebrities. As we mentioned in the limitations section (Page 18), we are actively working to increase the diversity of knowledge sources in RWKU. Following your suggestion, we have constructed **RWKU-Event** dataset containing 50 historical events as unlearning targets, including socially detrimental events such as terrorist attacks, massacres, murders, and political scandals. We have conducted experiments on this dataset, and as shown in Table 1, **the results aligned with those from the famous people dataset, further confirming the generalizability of our findings**. In the future, we plan to continuously expand the diversity of knowledge in RWKU, including the incorporation of scientific concepts. We believe expanding the diversity of unlearning targets enhances RWKU’s comprehensiveness, enabling it to address a broader range of unlearning scenarios and significantly increasing its value in real-world applications.
>
>
>
>
> | **Method**  | Forget FB $\downarrow$ | Forget QA $\downarrow$ | Forget AA $\downarrow$ | Forget All $\downarrow$ | Neighbor FB $\uparrow$ | Neighbor QA $\uparrow$ | Neighbor All $\uparrow$ | FM $\uparrow$ | RM $\downarrow$ | Gen $\uparrow$ | Rea $\uparrow$ | Tru $\uparrow$ | Fac $\uparrow$ | Flu $\uparrow$ |
> |-------------|-----|-----|-----|-----|-----|-----|-----|---------------|-----------------|-----|-----|-----|-----|-----|
> | Before | 94.1 | 82.8 | 83.1 | 85.8 | 95.9 | 81.9 | 89.0 | 255.8 | 260.0 | 65.1 | 42.4 | 37.0 | 54.5 | 711.7 |
> | ICU | 46.6 | **22.3** | **25.0** | **29.6** | 76.5 | 62.6 | 69.6 | 270.1 | 286.2 | 63.6 | 39.9 | **37.8** | 50.7 | 703.4 |
> | GA* | **39.1** | 34.2 | 37.9 | 37.6 | 72.2 | 59.4 | 65.9 | **1610.7** | 827.4 | 64.9 | 38.4 | **37.8** | 37.0 | 698.2 |
> | GA | 47.3 | 44.4 | 44.7 | 44.9 | 84.0 | **74.2** | **79.2** | 344.1 | 265.3 | 64.9 | 41.5 | 36.8 | 55.5 | 706.8 |
> | DPO | 58.8 | 42.3 | 46.9 | 49.2 | 66.4 | 54.1 | 60.3 | 274.4 | 274.4 | 63.2 | 40.6 | 33.3 | 15.9 | **722.7** |
> | NPO | 46.7 | 34.2 | 33.0 | 36.4 | 80.5 | 69.1 | 74.8 | 362.4 | 343.2 | 64.2 | **41.7** | 35.5 | 40.1 | 715.7 |
> | RT | 77.6 | 32.0 | 38.9 | 47.4 | **89.7** | 61.5 | 75.8 | 251.5 | **257.0** | **65.0** | 40.9 | 34.4 | **61.0** | 651.0 |
>
> **Table 1:** Results of LLaMA3-Instruct (8B) on RWKU-Event. The best results are highlighted in **bold**. * denotes the method trained on the pseudo ground truth forget corpus. $\uparrow$ means higher is better, and $\downarrow$ means lower is better.
>
>
> > Two pieces of work [1, 2] related to knowledge forgetting/unlearning were omitted.
>
> We sincerely appreciate you pointing this out. We will include references to these works in the next version of our paper to ensure a more comprehensive discussion.
>
> [1] Zheyuan Liu, Guangyao Dou, Zhaoxuan Tan, Yijun Tian, Meng Jiang. Towards Safer Large Language Models through Machine Unlearning.
>
> [2] Shiwen Ni, Dingwei Chen, Chengming Li, Xiping Hu, Ruifeng Xu, Min Yang. Forgetting before Learning: Utilizing Parametric Arithmetic for Knowledge Updating in Large Language Models.

---

> ### Author Response · Authors · 2024-08-26
> **A gentle reminder for discussion**
>
> Dear Reviewer,
>
> We would like to express our sincere gratitude for your great efforts and valuable comments. We have carefully addressed the concerns raised in your review and submitted a detailed rebuttal. As the discussion phase is about to close, we are eager to know if these responses adequately resolve the issues you highlighted. If you believe there is potential for improving the original rating, we are fully willing to make any additional efforts necessary. Please feel free to reach out with any further questions or concerns.
>
> Best regards,
>
> Authors

---

### Official Review · Reviewer_kTbb · 2024-07-22
**A benchmark for knowledge unlearning in LLMs (with three factors as design considerations)**

**Rating:** 7
**Confidence:** 3

**Review:**

See the "Strengths" below and other fields (for improvements/weaknesses).

**Strengths:**

+ The paper addresses a critical issue of unlearning in LLMs, which is essential for privacy and compliance with regulations like GDPR.

+ A comprehensive benchmark (RWKU) is introduced, featuring real-world knowledge and adversarial attack probes, making the evaluation realistic.

+ The experimental setup is thorough, covering various unlearning scenarios, models, and baseline methods, providing detailed insights into the efficacy and challenges of different approaches.

**Additional Feedback:**

Nil

**Clarity:**

The paper is generally well-written, with a clear structure that guides the reader through the motivation, methodology, experimental design, and results.

**Correctness:**

The claims in this work are basically correct given the the summary and details. The authors present a well-defined problem, justify the need for a benchmark, and propose a comprehensive evaluation framework. The construction of the RWKU benchmark seems sound, focusing on real-world knowledge unlearning with clear criteria for the forget set and retain set. The experimental design includes multiple models and baseline methods, which is appropriate for testing the effectiveness of unlearning techniques.

**Documentation:**

The authors provided the artifacts (including a publicly available link to their code repository) for reproducibility.

**Limitations:**

See the above "Opportunities For Improvement."

**Opportunities For Improvement:**

- The reliance on real-world famous people may introduce bias, as the unlearning targets are not representative of all types of knowledge within LLMs. This narrow scope does not encompass other types of knowledge (e.g., scientific concepts and historical events) that might also need to be unlearned. As a result, the generalizability of the benchmark to other knowledge types is limited.

- Some unlearning methods, particularly those that involve fine-tuning, may not be feasible for all models due to computational resource constraints.

- Additionally, the effectiveness heavily relies on adversarial probes, which might not comprehensively represent all possible ways forgotten knowledge could be extracted. This means that some unlearning methods may appear more effective than they are when evaluated in real-world scenarios where different or more sophisticated probing techniques might be used.

- The impact of unlearning on the overall performance of LLMs is not thoroughly discussed, particularly in terms of potential degradation in unrelated areas.

- There may be a risk that methods developed using this benchmark might overfit to the specific types of knowledge and adversarial probes included, rather than improving general unlearning capabilities.

**Relation To Prior Work:**

The paper clearly discusses how it differs from and builds upon previous contributions in the field of machine unlearning and LLM evaluation. The authors compare RWKU to existing benchmarks such as the "Who is Harry Potter?" task and the Weapons of Mass Destruction Proxy Benchmark (WMDP), highlighting the unique aspects of their benchmark, such as the focus on real-world knowledge and the inclusion of adversarial probes.

**Summary And Contributions:**

The paper introduces RWKU, a real-world knowledge unlearning benchmark for unlearning specific knowledge in large language models (LLMs). The authors identify the need to remove sensitive, copyrighted, or harmful knowledge from LLMs, addressing concerns related to privacy and regulatory compliance. RWKU targets unlearning real-world knowledge about 200 famous people and includes rigorous evaluations using membership inference attacks and adversarial probes. Extensive experiments across different unlearning scenarios, models, and baseline methods are conducted to provide insights and highlight the challenges in this field.

---

> ### Author Rebuttal · Authors · 2024-08-15
>
> Thank you for your careful and insightful reviews. We are greatly encouraged by your recognition that the inclusion of real-world knowledge and adversarial attack probes enhances the realism of our evaluation. We sincerely hope that our response addresses your concerns.
>
> > The reliance on real-world famous people may introduce bias, as the unlearning targets are not representative of all types of knowledge within LLMs. This narrow scope does not encompass other types of knowledge (e.g., scientific concepts and historical events) that might also need to be unlearned. As a result, the generalizability of the benchmark to other knowledge types is limited.
>
> We completely agree with your suggestion that using only famous people as unlearning targets might introduce bias. As we mentioned in the limitations section (Page 18), we are actively working to increase the diversity of knowledge sources in RWKU. Following your suggestion, we have constructed **RWKU-Event** dataset containing 50 historical events as unlearning targets, including socially detrimental events such as terrorist attacks, massacres, murders, and political scandals. We have conducted experiments on this dataset, and as shown in Table 1, **the results aligned with those from the famous people dataset, further confirming the generalizability of our findings**. In the future, we plan to continuously expand the diversity of knowledge in RWKU, including the incorporation of scientific concepts.
>
> | **Method**  | Forget FB $\downarrow$ | Forget QA $\downarrow$ | Forget AA $\downarrow$ | Forget All $\downarrow$ | Neighbor FB $\uparrow$ | Neighbor QA $\uparrow$ | Neighbor All $\uparrow$ | FM $\uparrow$ | RM $\downarrow$ | Gen $\uparrow$ | Rea $\uparrow$ | Tru $\uparrow$ | Fac $\uparrow$ | Flu $\uparrow$ |
> |-------------|-----|-----|-----|-----|-----|-----|-----|---------------|-----------------|-----|-----|-----|-----|-----|
> | Before | 94.1 | 82.8 | 83.1 | 85.8 | 95.9 | 81.9 | 89.0 | 255.8 | 260.0 | 65.1 | 42.4 | 37.0 | 54.5 | 711.7 |
> | ICU | 46.6 | **22.3** | **25.0** | **29.6** | 76.5 | 62.6 | 69.6 | 270.1 | 286.2 | 63.6 | 39.9 | **37.8** | 50.7 | 703.4 |
> | GA* | **39.1** | 34.2 | 37.9 | 37.6 | 72.2 | 59.4 | 65.9 | **1610.7** | 827.4 | 64.9 | 38.4 | **37.8** | 37.0 | 698.2 |
> | GA | 47.3 | 44.4 | 44.7 | 44.9 | 84.0 | **74.2** | **79.2** | 344.1 | 265.3 | 64.9 | 41.5 | 36.8 | 55.5 | 706.8 |
> | DPO | 58.8 | 42.3 | 46.9 | 49.2 | 66.4 | 54.1 | 60.3 | 274.4 | 274.4 | 63.2 | 40.6 | 33.3 | 15.9 | **722.7** |
> | NPO | 46.7 | 34.2 | 33.0 | 36.4 | 80.5 | 69.1 | 74.8 | 362.4 | 343.2 | 64.2 | **41.7** | 35.5 | 40.1 | 715.7 |
> | RT | 77.6 | 32.0 | 38.9 | 47.4 | **89.7** | 61.5 | 75.8 | 251.5 | **257.0** | **65.0** | 40.9 | 34.4 | **61.0** | 651.0 |
>
> **Table 1:** Results of LLaMA3-Instruct (8B) on **RWKU-Event**. The best results are highlighted in **bold**. * denotes the method trained on the pseudo ground truth forget corpus. $\uparrow$ means higher is better, and $\downarrow$ means lower is better.
>
> > Some unlearning methods, particularly those that involve fine-tuning, may not be feasible for all models due to computational resource constraints.
>
> We completely agree with your point regarding the feasibility of fine-tuning methods, particularly in terms of computational resource constraints. Indeed, the mainstream unlearning approaches currently involve gradient ascent on the data that needs to be forgotten, which can be resource-intensive.
>
> To address this, **we have experimented with various fine-tuning methods in our paper. In addition to full fine-tuning, we also explored partial-layer fine-tuning and LORA fine-tuning**. We find that fine-tuning the first few layers of the model can achieve comparable, or even better unlearning performance than full fine-tuning. On the other hand, our experiments also show that LoRA unlearns less on the forget set and forgets less on the retain set.
> Additionally, we have explored in-context unlearning and representation engineering as **parameter-free methods for unlearning**, both of which yielded promising results. These experimental results suggest that current unlearning methods still have considerable room for improvement in concept-level unlearning, motivating us to develop more **efficient** and **effective** unlearning techniques in the future.
>
> > Additionally, the effectiveness heavily relies on adversarial probes, which might not comprehensively represent all possible ways forgotten knowledge could be extracted. This means that some unlearning methods may appear more effective than they are when evaluated in real-world scenarios where different or more sophisticated probing techniques might be used.
>
> We want to highlight that, in constructing our benchmark, we deliberately avoid the potential bias of using only one type of probe. To ensure a comprehensive evaluation of unlearning effectiveness, **we meticulously develop four different probing methods: knowledge memorization, knowledge manipulation, adversarial attack, and membership inference attack**.
>
> Specifically, we use fill-in-the-blank probes to assess knowledge memorization and question-answer probes to evaluate knowledge manipulation. Additionally, we carefully design nine types of adversarial-attack probes, including prefix injection, affirmative suffix, role playing, reverse query, and others. Moreover, we also provide four membership inference attack methods to rigorously test unlearning efficacy. This diverse set of evaluation methods is intended to better represent the various ways in which forgotten knowledge might be extracted in real-world scenarios. By using these different probes, we aim to provide a more accurate and thorough evaluation of the effectiveness of unlearning methods.
>
> As shown in Table 1 and Figure 4 of our paper, different unlearning methods perform variably across these different probe types, highlighting the importance of using a comprehensive set of evaluations to accurately assess the effectiveness of each method.

---

> > ### Author Rebuttal · Authors · 2024-08-15
> >
> > > The impact of unlearning on the overall performance of LLMs is not thoroughly discussed, particularly in terms of potential degradation in unrelated areas.
> >
> >
> > Thank you for raising this concern. **We have indeed already conducted detailed experiments and discussions on the side effects on the model's overall capabilities**. We have conducted the retain assessment from two perspectives: (1) **Locality**: The unlearning process should be precise, without exceeding the boundaries of the target knowledge and perturbing the surrounding neighboring knowledge. (2) **Model Utility**: Beyond neighboring knowledge, the model’s performance on various real-world applications should not be impacted. We show the trade-off between unlearning efficacy, locality and model utility in Figure 3 of our paper.
> >
> >
> > - For locality, we define neighboring knowledge in the unlearning task as that which is closely related to, but not entirely contained within the scope of the unlearning targets. We design a neighbor set to test the impact of neighbor perturbation. And our experimental results show that it is challenging to balance the unlearning efficacy and locality. While unlearning the target knowledge, there are also side effects on neighboring knowledge.
> >
> > - For model utility, we assess the model utility on various capabilities, including general ability (MMLU [1]), reasoning ability (BBH [2]), truthfulness (TruthfulQA [3]), factuality (TriviaQA [4]), and fluency (AlpacaEval [5]). And we find that unlearning will encourage the model to generate hallucinations, thereby significantly affecting factuality and truthfulness.
> >
> > [1] Dan Hendrycks, Collin Burns, Steven Basart, Andy Zou, Mantas Mazeika, Dawn Song, Jacob Steinhardt. Measuring Massive Multitask Language Understanding. ICLR 2021.
> >
> > [2] Mirac Suzgun, Nathan Scales, Nathanael Schärli, Sebastian Gehrmann, Yi Tay, Hyung Won Chung, Aakanksha Chowdhery, Quoc V. Le, Ed H. Chi, Denny Zhou, Jason Wei. Challenging BIG-Bench Tasks and Whether Chain-of-Thought Can Solve Them. ACL (Findings) 2023.
> >
> > [3] Stephanie Lin, Jacob Hilton, Owain Evans. TruthfulQA: Measuring How Models Mimic Human Falsehoods. ACL 2022.
> >
> > [4] Mandar Joshi, Eunsol Choi, Daniel S. Weld, Luke Zettlemoyer. TriviaQA: A Large Scale Distantly Supervised Challenge Dataset for Reading Comprehension. ACL 2017.
> >
> > [5] Xuechen Li, Tianyi Zhang,Yann Dubois, Rohan Taori, Ishaan Gulrajani, Carlos Guestrin, Percy Liang, Tatsunori B. Hashimoto. AlpacaEval: An Automatic Evaluator of Instruction-following Models.
> >
> > > There may be a risk that methods developed using this benchmark might overfit to the specific types of knowledge and adversarial probes included, rather than improving general unlearning capabilities.
> >
> > As we mentioned earlier, we have incorporated historical events as unlearning targets based on your suggestion, and the experiments show that the results align with those from the famous people dataset, further confirming the generalizability of our findings. Additionally, our benchmark employs various methods to evaluate unlearning effectiveness, aiming to cover as many real-world knowledge extraction scenarios as possible. This ensures that the unlearning methods developed using our benchmark have strong generalization capabilities.

---

> ### Author Response · Authors · 2024-08-26
> **A gentle reminder for discussion**
>
> Dear Reviewer,
>
> We would like to express our sincere gratitude for your great efforts and valuable comments. We have carefully addressed the concerns raised in your review and submitted a detailed rebuttal. As the discussion phase is about to close, we are eager to know if these responses adequately resolve the issues you highlighted. If you believe there is potential for improving the original rating, we are fully willing to make any additional efforts necessary. Please feel free to reach out with any further questions or concerns.
>
> Best regards,
>
> Authors

---

> > ### Comment · Reviewer_kTbb · 2024-08-27
> >
> > Thanks for clarifying/addressing my concerns and supplementing new results.
> > I am still positive about this paper.

---

> > > ### Author Response · Authors · 2024-08-27
> > > **Appreciate your new response and updated score**
> > >
> > > Dear Reviewer,
> > >
> > > Thank you so much for your careful and insightful feedback and for raising the score! We are glad that our responses addressed your concerns. Your advice is invaluable and has provided us with guidance towards a more comprehensive paper. We truly appreciate your support for our work.

---

### Official Review · Reviewer_nFJQ · 2024-07-24
**Solid work with design flaws**

**Rating:** 4
**Confidence:** 4

**Review:**

Pros:

1. This paper adopts some common evaluation practices in knowledge editing (like considering both efficacy and locality, as well as the batching setting) and utilizes adversarial attack techniques in evaluation. I think these practices are helpful in comprehensively evaluating knowledge unlearning and should be followed by future works.

2. The experiments are solid, covering a wide range of LLMs, unlearning methods, and settings, which should be valuable for the follow-up works.

Cons:

1. My major concern is that I do not find the new task setting and knowledge source reasonable, which are two core contributions of this paper. For the new task setting that does knowledge unlearning without providing data to be unlearned, the authors justify it as providing sensitive information may cause "secondary leakage", which I think the regulators instead of unlearning techniques should be blamed if it happens. Without the exact data to be unlearned, the objective of unlearning is very vague and I don't think it can be correctly evaluated. To say the least, we don't have this requirement even for non-black-box systems (like filing a DMCA complaint to a website host). Why should we expect this in LLMs? For the knowledge source, I appreciate the authors' efforts in collecting and cleansing the data, but I think the objective to forget all the information about an entity (specifically, famous people) instead of specific attributes of an entity (like social security number of a person) is too high and far from realistic needs.

2. All the forget probes are generated with GPT-4. (1) The specific API endpoints should be provided for reproducibility. (2) The data should undergo human checks to ensure its quality and diversity, and the human evaluation results should be provided to help understand the utility of the benchmark.

**Strengths:**

Please refer to the pros above.

**Additional Feedback:**

Line 13: duplicate "four"

**Clarity:**

In general, the paper is clear. I would enjoy the paper better if its expressions could be more informative rather than conceptual.

**Correctness:**

I have major concerns regarding the benchmark design. Please refer to the cons above.

**Documentation:**

Yes.

**Ethics:**

No.

**Limitations:**

Yes

**Opportunities For Improvement:**

Please refer to the cons above.

**Relation To Prior Work:**

Yes.

**Summary And Contributions:**

This paper proposes a new benchmark of knowledge unlearning for large language models, with data collected from Wikipedia and new evaluation designs. It reformulates the machine unlearning problem into a new setting without providing forget and retain corpora. The evaluation design combines practices from knowledge editing and adversarial attack research.

---

> ### Author Rebuttal · Authors · 2024-08-15
>
> Thanks for your careful and insightful reviews. We are deeply encouraged that you consider our paper to be valuable for follow-up works. We sincerely hope that our response addresses your concerns.
>
> > My major concern is that I do not find the new task setting and knowledge source reasonable, which are two core contributions of this paper.
>
> Regarding the task setting and knowledge source, it's important to consider that our work focuses on **concept-level unlearning**, which is distinct from traditional **instance-level unlearning**. Instance-level unlearning is relatively straightforward, often involving techniques like gradient ascent on specific training data points that need to be forgotten, such as a person's social security number. In contrast, our study focuses on concept-level forgetting, which is significantly more challenging as it requires the removal of all knowledge associated with a particular concept from the model, rather than deleting single data points. Importantly, **our unlearning targets for concept-level forgetting are clearly defined, ensuring that the boundaries of the knowledge to be forgotten are precise**.
>
> For the **task setting**, we acknowledge your point that secondary leakage of sensitive information could be attributed to regulatory bodies rather than unlearning techniques. While confidentiality agreements might ensure data security, we also recognize the inherent difficulty in collecting data for unlearning in our paper (Lines 46-48). During a model's pre-training process, **knowledge about a particular concept can be derived from multiple training data points, making it nearly impossible to identify all relevant data, much like searching for a needle in a haystack**.
>
> For example, consider the challenge of removing the concept of "Harry Potter" from a model. Simply unlearning the Harry Potter series of books is insufficient [1, 2], as the model likely acquired knowledge from numerous other sources, such as blogs, social media discussions, and even indirect mentions in websites. This widespread dissemination makes it impossible to exhaust all relevant training data points, highlighting the complexity of achieving comprehensive concept-level unlearning.
>
> Our experimental results further emphasize this point, demonstrating that unlearning using only Wikipedia snippets is not fully effective. For the reasons mentioned above, while we do not provide a dedicated forget corpus, we do offer Wikipedia snippets for comparison. More importantly, our task setting does not exclude the use of forget corpus. For example, relevant data can be obtained through techniques such as retrieval or influence estimation, further encouraging the development of new data attribution methods. Additionally, **our task setting also encourages the development of more data-efficient and training-efficient methods for concept-level unlearning**.
>
> For the **knowledge source**, as mentioned earlier, we focus on concept-level unlearning, which is not only inherently more difficult but also significantly different from instance-level unlearning. We acknowledge that this is indeed a high goal, however, we believe concept-level unlearning is also a very realistic and worthwhile problem to investigate. For instance, **there may be situations where it’s necessary to remove all knowledge related to a harmful or biased concept, a specific historical event, or a sensitive entity to ensure compliance with ethical standards or privacy regulations**. Additionally, we have constructed a dataset containing 50 historical events as unlearning targets, including socially detrimental events such as terrorist attacks, massacres, murders, and political scandals. Recently, there have also been some concurrent works focusing on concept-level unlearning [3, 4, 5].
>
> Furthermore, our experimental results in Section 5 show that **existing unlearning methods fall short when directly applied to concept-level unlearning and often lead to undesirable side effects on neighboring knowledge**. This highlights the need for more precise concept-level unlearning techniques in the future, potentially requiring us to move beyond the current gradient ascent-based paradigm.
>
> [1] Adam Shostack. The Boy Who Survived: Removing Harry Potter from an LLM is harder than reported.
>
> [2] Aengus Lynch, Phillip Guo, Aidan Ewart, Stephen Casper, Dylan Hadfield-Menell. Eight Methods to Evaluate Robust Unlearning in LLMs.
>
> [3] Yihuai Hong, Lei Yu, Shauli Ravfogel, Haiqin Yang, Mor Geva. Intrinsic Evaluation of Unlearning Using Parametric Knowledge Traces.
>
> [4] Weitao Ma, Xiaocheng Feng, Weihong Zhong, Lei Huang, Yangfan Ye, Bing Qin. Rethinking Entity-level Unlearning for Large Language Models.
>
> [5] Yujian Liu, Yang Zhang, Tommi Jaakkola, Shiyu Chang. Revisiting Who's Harry Potter: Towards Targeted Unlearning from a Causal Intervention Perspective.

---

> > ### Author Rebuttal · Authors · 2024-08-15
> >
> > > The specific API endpoints should be provided for reproducibility.
> >
> > We apologize for not specifying the exact API endpoints used in the paper. We used **GPT-4 Turbo** when constructing the benchmark.
> >
> > > The data should undergo human checks to ensure its quality and diversity, and the human evaluation results should be provided to help understand the utility of the benchmark.
> >
> > Thank you for pointing out the importance of assessing the quality of the probes. Based on your suggestion, we evaluate the probes from the perspectives of diversity and correctness.
> >
> > (1) For the **diversity**, we measure semantic diversity by calculating the similarity between the probe embeddings. Specifically, we use the [all-MiniLM-L6-v2](https://huggingface.co/sentence-transformers/all-MiniLM-L6-v2) to map probes to a dense vector space. Given that RWKU is a dataset with multiple unlearning targets, we calculate two types of semantic similarity:
> >
> > - **Intra-Target Similarity**: The similarity between all queries within a single target.
> > - **Cross-Target Similarity**: The similarity between queries across multiple targets.
> >
> > For these two metrics, lower scores indicate greater probe diversity. As a comparison, we select two other datasets that also contain multiple unlearning targets: TOFU [6] and CONCEPTVECTORS [3]. As shown in Table 1, RWKU demonstrates strong probe diversity in both Intra-Target and Cross-Target similarity.
> >
> >
> > | **Dataset**        | **Intra-Target Similarity** $\downarrow$ | **Cross-Target Similarity** $\downarrow$ |
> > |----------------|--------------|--------------|
> > | TOFU           | 61.9         | 13.5         |
> > | CONCEPTVECTORS | 48.3         | **9.4**          |
> > | **RWKU**           | **45.1**         | 13.3         |
> >
> > **Table 1:** Statistics of Probe Diversity.
> >
> >
> > (2) For the **correctness**, we maintain strict quality control over the generated probes during the construction of our benchmark. As shown in Figure 1 of the supplementary material, we provide the workflow of dataset construction. First, we retrieve fragments related to the unlearning targets from Wikipedia and use GPT-4 to generate an excess of query-answer pairs based on these fragments, leveraging an RAG approach to significantly reduce the model's hallucinations.
> >
> >
> > Then, we use LLaMA3-Instruct (8B) to filter the generated probes, retaining only those probes that the model could answer correctly, while further reducing the inconsistencies between the questions and answers. During this process, we filter out **57.1%** of the probes. Finally, we manually check these probes to ensure their format and type are correct, discarding an additional **3.7%** of the probes in this final stage.
> >
> > Based on your suggestion, we have conducted a random sampling evaluation of 1,000 probes to assess their accuracy. The evaluation by GPT-4 showed an accuracy rate of **98.7%**, while the manual evaluation achieved an accuracy rate of **99.1%**, demonstrating the high quality of the probes we constructed.
> >
> > [6] Pratyush Maini, Zhili Feng, Avi Schwarzschild, Zachary C. Lipton, J. Zico Kolter. TOFU: A Task of Fictitious Unlearning for LLMs.

---

> ### Author Response · Authors · 2024-08-26
> **A gentle reminder for discussion**
>
> Dear Reviewer,
>
> We would like to express our sincere gratitude for your great efforts and valuable comments. We have carefully addressed the concerns raised in your review and submitted a detailed rebuttal. As the discussion phase is about to close, we are eager to know if these responses adequately resolve the issues you highlighted. If you believe there is potential for improving the original rating, we are fully willing to make any additional efforts necessary. Please feel free to reach out with any further questions or concerns.
>
> Best regards,
>
> Authors

---

> ### Author Response · Authors · 2024-08-28
> **Adding a qualitative diversity analysis**
>
> Dear Reviewer nFJQ,
>
> To gain a deeper understanding of RWKU, we recently incorporated a manual qualitative analysis of probe diversity based on Reviewer 7nBB's suggestions. Specially, we performed a manual clustering analysis on 200 randomly sampled probes. Through manual analysis, **we identified 48 distinct cluster centers, demonstrating that the questions cover a wide range of topics**. Additionally, we have listed the top 15 manually annotated cluster categories in Table 1, which are ranked based on the number of probes contained within each category. We can observe that GPT-4 shows a certain preference when generating probes, tending to produce questions related to 'Received awards', 'Acted films', 'Played roles', 'Created works' and 'Aliases'.
>
> | Category                 | Count |
> |--------------------------|-------|
> | Received awards           | 12%    |
> | Acted films               | 8.5%    |
> | Played roles              | 7%    |
> | Created works             | 5.5%    |
> | Aliases                   | 5%    |
> | Released albums           | 4.5%     |
> | Participated shows        | 3.5%     |
> | Established organizations | 3%     |
> | Family members            | 3%     |
> | Debut                     | 3%     |
> | Affiliated team           | 2.5%     |
> | Place of birth            | 2.5%     |
> | High school               | 2.5%     |
> | Date of birth             | 2%     |
> | University                | 2%     |
>
> **Table 1:** Statistics of Cluster Categories.
>
> Furthermore, we also analyzed the lexical similarity of questions within each category. Our findings show that **for most categories, the questions are indeed diverse**. For instance, within the 'Received Awards' category, the probes demonstrate a diverse range of questioning styles, including:
>
> - What X did Y win for Z?
> - What award was X nominated for his performance in Y?
> - Can you tell me which award X received in Y for Z?
> - Can you name the player who won X?
> - For which film did X win Y for Z?
> - Who won X for his roles in Y?
> - What year did X receive Y?
> - Which of the following awards has X won: A, B, C?
> - Which of the following awards was X nominated for due to his role in Y: A, B, C?
> - I noticed X won Y. Can you tell me for which film she won this award?
> - ...
>
> Similarly, within the 'Debut' category, the probes also take on various forms of questioning, including:
>
> - Can you let me know the year X made his on-screen feature film debut?
> - Could you tell me the film that marked X's debut?
> - Can you tell me in which movie X made his film debut?
> - Which cinematic masterpiece marked X's significant on-screen debut?
> - Which film marked X's debut?
> - Could you tell me what the debut album of X from his solo career was?
> - ...
>
> However, **for certain categories, such as 'Place of birth' and 'Date of birth,' GPT-4 tends to follow specific templates**. For example, in the 'Place of Birth' category, the generated probes often take forms like 'In which city was X born?' or 'What city was X born in?' Similarly, in the 'Date of Birth' category, the questions frequently follow the pattern 'What year was X born?'. We believe this may be due to the inherently straightforward nature of such questions, which often leads to more uniform phrasing.
>
> Through manual clustering analysis, we have demonstrated that the probes exhibit a high degree of diversity across various question categories, and for most categories, the questions demonstrate a wide range of questioning styles. However, for certain specific categories, GPT-4 tends to follow a limited set of templates. This observation highlights the need to focus more on these categories in our future work, where we will actively employ paraphrasing strategies to enhance their diversity.
>
> **We sincerely hope this additional experiment will contribute to a better understanding of probe diversity and help address your concerns. As we approach the end of the discussion phase, we remain fully committed to refining our work and would greatly appreciate any additional feedback you may have.**

---

> ### Author Response · Authors · 2024-08-30
> **Looking forward to your feedback**
>
> Dear Reviewer nFJQ,
>
> We would like to once again express our gratitude for taking the time to review our paper and consider our rebuttal. **In response to your feedback, we have clarified the motivation for the new task setting and knowledge sources, and we have manually evaluated the quality of the constructed probes**. As the discussion phase will end on August 31st, we kindly ask if our response has satisfactorily addressed all of your concerns. We sincerely hope to engage in a discussion with you to improve our initial rating.
>
> Best regards,
>
> Authors

---

### Official Review · Reviewer_7nBB · 2024-07-26
**Thorough Empirical Contribution with Some Details Requiring Clarification**

**Rating:** 7
**Confidence:** 3
**Clarity:** The paper is very well-written

**Review:**

This paper has several interesting insights, and the empirical contributions are extensive. I appreciated many aspects of this work, such as the variety of ways the work tests for true unlearning. At the same time, there are some missing details which left me unable to get a full picture of how to interpret the results of the probes (see opportunities for improvement).

**Strengths:**

This paper has some interesting findings about unlearning knowledge, such as/;
- Compared to QA-style probes, adversarial and cloze probes are effective at extracting “forgotten” knowledge
- Tradeoff between unlearning efficacy and locality (as measured by probing neighboring knowledge)
- Batch target unlearning being more difficult than single target, and leding to model collapse
- Gradient ascent performs well for knowledge unlearning.

**Additional Feedback:**

N/A

**Correctness:**

There are some missing quality assessments on some parts of the resulting dataset.

**Documentation:**

Yes

**Limitations:**

The paper did not discuss negative societal impact. I imagine this is worthy of a discussion, specifically when focusing on unlearning real-world entities.

**Opportunities For Improvement:**

- It would be good to discuss how well methods that do well on RWKU, will do on long-tail entities.
- I am not sure how realistic the benchmark setting is (one of the central claims is that this benchmark is practical to real-world scenarios). However, it remains unclear if there are real-world scenarios requiring a model to forget all its knowledge about a particular entity,
- The work is missing quality assessment of the probes (for example, how diverse/non-overlapping is the question set, how correct are the questions and answers).
- It would be good to describe how the “knowledge points” for fill-in-the-blank style probes were selected. Specifically, I'm concerned some of these knowledge points are nouns or terms that are easy to guess, and may not require knowledge of the particular entity. I would appreciate clarification on how this was controlled for.

EDIT: I Have read the author response and appreciate the additional experiments done by the authors. I have raised my score.

**Relation To Prior Work:**

The paper adequately compares to related work

**Summary And Contributions:**

This paper contributes the RWKU benchmark to measure model’s ability to unlearn knowledge. The benchmark examines the unlearning setting where : (1) neither a forget corpus or a retain corpus is available, (2) unlearning target is knowledge about 200 famous people (3) evaluation is focused on model capabilities as a measure of utility. To measure unlearning performance, four MIA methods and nine probes are used (ideally an MIA method should not successfully be able to identify that a model was trained on the unlearn target for effective unlearning).

---

> ### Author Rebuttal · Authors · 2024-08-15
>
> Thanks for your careful and insightful reviews. We are deeply encouraged that you recognized and appreciated several aspects of our work. We would also like to address the missing details to help you gain a more complete understanding of our research.
>
>
>
> > It would be good to discuss how well methods that do well on RWKU, will do on long-tail entities.
>
> Thank you for your insightful suggestion. We select popular entities as unlearning targets because this knowledge is widely present across various large language models, enhancing the broad applicability of our benchmark. Given the challenges large language models struggle to learn long-tail knowledge [1], we select 20 mid-popularity entities as unlearning targets and create the **RWKU-Tail** dataset using the same workflow. We also discard those probes that the original model (i.e., LLaMA3-Instruct (8B)) can not answer correctly. Considering that models grasp popular and long-tail knowledge differently, we uniformly use the corresponding Wikipedia pages as the unlearning corpus. As shown in Tables 1 and 2, for long-tail knowledge, NPO causes a significant 76.9% performance drop across all forgetting probes, while the performance on all neighbor probes only decreases slightly by 5.9%. However, for popular knowledge in RWKU, NPO causes a slight 38.7% performance drop across all forgetting probes, while the performance on all neighbor probes decreases significantly by 19.7%. We can observe that **unlearning long-tail knowledge is easier and has a smaller impact on neighbor knowledge**. This highlights that **unlearning popular knowledge in RWKU is more challenging, requiring the development of more effective unlearning methods to minimize side effects**.
>
>
>
>
>
>
>
>
>
> | **Method**  | Forget FB $\downarrow$ | Forget QA $\downarrow$ | Forget AA $\downarrow$ | Forget All $\downarrow$ | Neighbor FB $\uparrow$ | Neighbor QA $\uparrow$ | Neighbor All $\uparrow$ | FM $\uparrow$ | RM $\downarrow$ | Gen $\uparrow$ | Rea $\uparrow$ | Tru $\uparrow$ | Fac $\uparrow$ | Flu $\uparrow$ |
> |-------------|-----|-----|-----|-----|-----|-----|-----|---------------|-----------------|-----|-----|-----|-----|-----|
> | **Before**  | 96.0 | 79.6 | 84.5 | 85.4 | 99.0 | 91.5 | 95.2 | 266.2 | 283.7 | 65.4 | 42.1 | 39.6 | 54.7 | 704.0 |
> | **GA**      | 32.2 | 22.9 | 27.7 | 29.2 | 84.8 | 81.9 | 83.3 | 2805.6 | 1346.3 | 65.0 | 39.9 | 40.0 | 40.1 | 693.3 |
> | **NPO**     | 30.4 | 11.4 | 15.5 | 19.7 | 93.5 | 85.6 | 89.6 | 635.0 | 370.1 | 65.6 | 40.1 | 41.3 | 39.4 | 717.6 |
>
>
> **Table 1:** Results of LLaMA3-Instruct (8B) on RWKU-Tail. $\uparrow$ means higher is better, and $\downarrow$ means lower is better.
>
> | **Method**  | Forget FB $\downarrow$ | Forget QA $\downarrow$ | Forget AA $\downarrow$ | Forget All $\downarrow$ | Neighbor FB $\uparrow$ | Neighbor QA $\uparrow$ | Neighbor All $\uparrow$ | FM $\uparrow$ | RM $\downarrow$ | Gen $\uparrow$ | Rea $\uparrow$ | Tru $\uparrow$ | Fac $\uparrow$ | Flu $\uparrow$ |
> |-------------|-----|-----|-----|-----|-----|-----|-----|---------------|-----------------|-----|-----|-----|-----|-----|
> | Before | 85.9         | 76.4| 77.7| 79.6| 95.6           | 85.3| 90.7| 226.7   | 230.4| 65.7          | 42.3| 36.8| 53.5| 705.8|
> | GA     | 40.7         | 36.5| 43.7| 41.4| 68.6           | 68.6| 68.1| 1640.9  | 766.2| 65.5          | 39.7| 37.8| 41.9| 692.4|
> | NPO    | 50.2         | 48.3| 47.9| 48.8| 78.4           | 68.7| 72.8| 461.4   | 321.6| 65.2          | 40.9| 37.4| 40.9| 713.6|
>
> **Table 2:** Results of LLaMA3-Instruct (8B) on RWKU-Popular. $\uparrow$ means higher is better, and $\downarrow$ means lower is better.
>
>
> [1] Nikhil Kandpal, Haikang Deng, Adam Roberts, Eric Wallace, Colin Raffel. Large Language Models Struggle to Learn Long-Tail Knowledge. ICML 2023.
>
> > It remains unclear if there are real-world scenarios requiring a model to forget all its knowledge about a particular entity.
>
>
>
> We understand your concerns, and we would like to emphasize that our work is distinct from previous work that focuses on instance-level unlearning, which is relatively simple and requires the model to forget specific training data points (e.g., a person’s social security number). In contrast, **our work focuses on concept-level unlearning, which is more challenging as it requires removing all knowledge associated with a particular concept from the model**. **Concept-level unlearning is not limited to particular entities; it can also be applied to forgetting historical events, harmful concepts, and other types of knowledge, making it highly valuable in real-world scenarios**. Additionally, we have constructed a dataset containing 50 historical events as unlearning targets, including socially detrimental events such as terrorist attacks, massacres, murders, and political scandals. We also conduct experiments on this dataset, and the results align with those from the famous people dataset, further confirming the generalizability of our findings. Recently, there have also been some concurrent works focusing on concept-level unlearning [2, 3, 4].
>
> Furthermore, our experimental results in Section 5 show that **existing unlearning methods fall short when directly applied to concept-level unlearning and often lead to undesirable side effects on neighboring knowledge**. This highlights the need for more precise concept-level unlearning techniques in the future, potentially requiring us to move beyond the current gradient ascent-based paradigm.
>
> [2] Yihuai Hong, Lei Yu, Shauli Ravfogel, Haiqin Yang, Mor Geva. Intrinsic Evaluation of Unlearning Using Parametric Knowledge Traces.
>
> [3] Weitao Ma, Xiaocheng Feng, Weihong Zhong, Lei Huang, Yangfan Ye, Bing Qin. Rethinking Entity-level Unlearning for Large Language Models.
>
> [4] Yujian Liu, Yang Zhang, Tommi Jaakkola, Shiyu Chang. Revisiting Who's Harry Potter: Towards Targeted Unlearning from a Causal Intervention Perspective.

---

> > ### Author Rebuttal · Authors · 2024-08-15
> >
> > > The work is missing quality assessment of the probes (for example, how diverse/non-overlapping is the question set, how correct are the questions and answers).
> >
> > Thank you for pointing out the importance of assessing the quality of the probes. Based on your suggestion, we evaluate the probes from the perspectives of diversity and correctness.
> >
> > (1) For the **diversity**, we measure semantic diversity by calculating the similarity between the probe embeddings. Specifically, we use the [all-MiniLM-L6-v2](https://huggingface.co/sentence-transformers/all-MiniLM-L6-v2) to map probes to a dense vector space. Given that RWKU is a dataset with multiple unlearning targets, we calculate two types of semantic similarity:
> >
> > - **Intra-Target Similarity**: The similarity between all queries within a single target.
> > - **Cross-Target Similarity**: The similarity between queries across multiple targets.
> >
> > For these two metrics, lower scores indicate greater probe diversity. As a comparison, we select two other datasets that also contain multiple unlearning targets: TOFU [5] and CONCEPTVECTORS [2]. As shown in Table 3, RWKU demonstrates strong probe diversity in both Intra-Target and Cross-Target similarity.
> >
> > | **Dataset**        | **Intra-Target Similarity** $\downarrow$ | **Cross-Target Similarity** $\downarrow$ |
> > |----------------|--------------|--------------|
> > | TOFU           | 61.9         | 13.5         |
> > | CONCEPTVECTORS | 48.3         | **9.4**          |
> > | **RWKU**           | **45.1**         | 13.3         |
> >
> > **Table 3:** Statistics of Probe Diversity.
> >
> > (2) For the **correctness**, we maintain strict quality control over the generated probes during the construction of our benchmark. As shown in Figure 1 of the supplementary material, we provide the workflow of dataset construction. First, we retrieve fragments related to the unlearning targets from Wikipedia and use GPT-4 to generate an excess of query-answer pairs based on these fragments, leveraging an RAG approach to significantly reduce the model's hallucinations.
> >
> > Then, we use LLaMA3-Instruct (8B) to filter the generated probes, retaining only those probes that the model could answer correctly, while further reducing the inconsistencies between the questions and answers. During this process, we filter out **57.1%** of the probes. Finally, we manually check these probes to ensure their format and type are correct, discarding an additional **3.7%** of the probes in this final stage.
> >
> > Based on your suggestion, we have conducted a random sampling evaluation of 1,000 probes to assess their accuracy. The evaluation by GPT-4 showed an accuracy rate of **98.7%**, while the manual evaluation achieved an accuracy rate of **99.1%**, demonstrating the high quality of the probes we constructed.
> >
> > [5] Pratyush Maini, Zhili Feng, Avi Schwarzschild, Zachary C. Lipton, J. Zico Kolter. TOFU: A Task of Fictitious Unlearning for LLMs.
> >
> > > It would be good to describe how the "knowledge points" for fill-in-the-blank style probes were selected.
> >
> > We apologize for not providing enough details on this matter. For the fill-in-the-blank style probes, we instruct GPT-4 to generate questions that are closely related to factual knowledge, while avoiding enumeration, open-ended, or non-factual questions. Additionally, we provide several crafted examples within the context to guide the generation process. Details of probe construction are available in Appendix C.
> >
> > We also adopt an NER toolkit in [Stanza](https://github.com/stanfordnlp/stanza) to classify the knowledge points. As shown in Table 4, **these knowledge points are meaningful entities rather than simple nouns, primarily including categories such as PERSON, MISC, DATE, GPE, CARDINAL**. Additionally, we test a relatively smaller model, Qwen2-0.5B-Instruct, which has sufficient linguistic capability to predict simple nouns or terms. However, due to its limited parameter size, it lacks the capacity to fully store substantial factual knowledge. As shown in Table 5, Qwen2-0.5B-Instruct performs poorly on these fill-in-the-blank style probes, indicating that **the model cannot simply guess the answers, especially when it does not possess the required knowledge of the particular entity**.
> >
> > | Knowledge Point Type   | Percentage |
> > |------------------------|------------|
> > | PERSON                 | 43.67%     |
> > | MISC                   | 28.65%     |
> > | DATE                   | 8.57%      |
> > | GPE                    | 6.54%      |
> > | CARDINAL               | 6.06%      |
> > | ORG                    | 3.91%      |
> > | NORP                   | 0.81%      |
> > | ORDINAL                | 0.64%      |
> > | WORK_OF_ART            | 0.36%      |
> > | EVENT                  | 0.25%      |
> > | FAC                    | 0.14%      |
> > | LANGUAGE               | 0.12%      |
> > | LOC                    | 0.12%      |
> > | TIME                   | 0.05%      |
> > | LAW                    | 0.05%      |
> > | MONEY                  | 0.03%      |
> > | PRODUCT                | 0.02%      |
> >
> > **Table 4:** Distribution of Knowledge Point Type.
> >
> > | **Method**  | Forget FB  | Forget QA  | Forget AA  | Forget All | Neighbor FB  | Neighbor QA  | Neighbor All
> > |-------------|-----|-----|-----|-----|-----|-----|-----|
> > | LLaMA3 | 85.9          | 76.4| 77.7| 79.6| 95.6            | 85.3| 90.7|
> > | Mistral| 64.0          | 52.1| 57.8| 58.2| 74.0            | 61.9| 68.4|
> > | LLaMA2 | 51.8          | 46.9| 57.5| 53.8| 63.7            | 64.6| 64.1|
> > | Phi-3  | 47.1          | 47.4| 55.8| 51.8| 56.2            | 61.4| 58.3|
> > | Qwen2  | 17.5          | 18.4| 22.5| 20.4| 21.9            | 18.7| 20.1|
> >
> > **Table 5:** Performance of different LLMs without unlearning, where Qwen2 denotes Qwen2-Instruct (0.5B), LLaMA3 denotes LLaMA3-Instruct (8B), Mistral denotes Mistral-Instruct-v0.2 (7B), LLaMA2 denotes LLaMA2-Chat (7B), and Phi-3 denotes Phi-3 Mini-4K-Instruct (3.8B).

---

> > > ### Author Rebuttal · Authors · 2024-08-15
> > >
> > > > The paper did not discuss negative societal impact. I imagine this is worthy of a discussion, specifically when focusing on unlearning real-world entities.
> > >
> > >
> > > Thank you for pointing this out. We acknowledge the importance of discussing the potential negative societal impacts of unlearning, especially when it involves real-world entities. It's important to recognize that while unlearning specific knowledge from models can be beneficial in certain contexts, it also carries significant risks. For example, the ability to selectively erase knowledge about real-world entities could be misused to intentionally remove critical information, potentially distorting the model's understanding of the world. **This becomes especially concerning if such capabilities are employed to manipulate information or obscure historical facts**.
> > > To mitigate these risks, it’s essential to establish strict guidelines and controls, similar to those used in preventing data poisoning and backdoor attacks. This includes ensuring that the **unlearning process is transparent, traceable, and overseen to prevent misuse**.

---

> > ### Comment · Reviewer_7nBB · 2024-08-26
> >
> > Thank you for engaging with the review. Could I request an alternative way of analyzing probe diversity: would it be possible to do a similar qualitative analysis of 100 or 1000 samples on the probes and manually cluster them based on similarity and report the results?
> >
> > For example you could quantify:
> >
> > (1) How repetitive are questions in terms of what they are asking (for example, does GPT-4 generate a lot of questions that are of the form "Where was X born?", causing a skew in the resulting probe dataset)
> >
> > (2) How similar lexically are the questions? (For example, does GPT-4 seem to generate questions following a few  templates?)

---

> > ### Author Response · Authors · 2024-08-27
> > **Adding a manual qualitative analysis**
> >
> > Dear Reviewer,
> >
> > Thank you very much for your insightful feedback. We greatly appreciate your suggestion, as it provides a valuable approach for evaluating the diversity of the probes. Your suggested analysis method has deeply inspired us.
> >
> > Based on your suggestion, we conduct a manual clustering analysis on 200 randomly sampled probes. Through manual analysis, **we identify 48 distinct cluster centers, demonstrating that the questions cover a wide range of topics**. Additionally, we list the top 15 manually annotated cluster categories in Table 1, which are ranked based on the number of probes contained within each category. We can observe that GPT-4 shows a certain preference when generating probes, tending to produce questions related to 'Received awards', 'Acted films', 'Played roles', 'Created works' and 'Aliases'.
> >
> > | Category                 | Percentage |
> > |--------------------------|-------|
> > | Received awards           | 12%    |
> > | Acted films               | 8.5%    |
> > | Played roles              | 7%    |
> > | Created works             | 5.5%    |
> > | Aliases                   | 5%    |
> > | Released albums           | 4.5%     |
> > | Participated shows        | 3.5%     |
> > | Established organizations | 3%     |
> > | Family members            | 3%     |
> > | Debut                     | 3%     |
> > | Affiliated team           | 2.5%     |
> > | Place of birth            | 2.5%     |
> > | High school               | 2.5%     |
> > | Date of birth             | 2%     |
> > | University                | 2%     |
> >
> > **Table 1:** Statistics of Cluster Categories.
> >
> > Furthermore, we analyze the lexical similarity of questions within each category. Our findings show that **for most categories, the questions are indeed diverse**. For instance, within the 'Received Awards' category, the probes demonstrate a diverse range of questioning styles, including:
> >
> > - What X did Y win for Z?
> > - What award was X nominated for his performance in Y?
> > - Can you tell me which award X received in Y for Z?
> > - Can you name the player who won X?
> > - For which film did X win Y for Z?
> > - Who won X for his roles in Y?
> > - What year did X receive Y?
> > - Which of the following awards has X won: A, B, C?
> > - Which of the following awards was X nominated for due to his role in Y: A, B, C?
> > - I noticed X won Y. Can you tell me for which film she won this award?
> > - ...
> >
> > Similarly, within the 'Debut' category, the probes also take on various forms of questioning, including:
> >
> > - Can you let me know the year X made his on-screen feature film debut?
> > - Could you tell me the film that marked X's debut?
> > - Can you tell me in which movie X made his film debut?
> > - Which cinematic masterpiece marked X's significant on-screen debut?
> > - Which film marked X's debut?
> > - Could you tell me what the debut album of X from his solo career was?
> > - ...
> >
> > However, **for certain categories, such as 'Place of birth' and 'Date of birth,' GPT-4 tends to follow specific templates**. For example, in the 'Place of Birth' category, the generated probes often take forms like 'In which city was X born?' or 'What city was X born in?' Similarly, in the 'Date of Birth' category, the questions frequently follow the pattern 'What year was X born?'. We believe this may be due to the inherently straightforward nature of such questions, which often leads to more uniform phrasing.
> >
> > We sincerely appreciate your constructive feedback on the analysis of probe diversity, as it contributes significantly to a deeper understanding of RWKU. Through manual clustering analysis, we have demonstrated that the probes exhibit a high degree of diversity across various question categories, and for most categories, the questions demonstrate a wide range of questioning styles. However, for certain specific categories, GPT-4 tends to follow a limited set of templates. This observation highlights the need to focus more on these categories in our future work, where we will actively employ paraphrasing strategies to enhance their diversity.

---

> ### Author Response · Authors · 2024-08-26
> **A gentle reminder for discussion**
>
> Dear Reviewer,
>
> We would like to express our sincere gratitude for your great efforts and valuable comments. We have carefully addressed the concerns raised in your review and submitted a detailed rebuttal. As the discussion phase is about to close, we are eager to know if these responses adequately resolve the issues you highlighted. If you believe there is potential for improving the original rating, we are fully willing to make any additional efforts necessary. Please feel free to reach out with any further questions or concerns.
>
> Best regards,
>
> Authors

---

> ### Comment · Reviewer_7nBB · 2024-08-27
>
> Thank you to the authors for their response, and for adding this analysis. I am leaning positive about this paper and have raised my score.

---

> > ### Author Response · Authors · 2024-08-27
> > **Appreciate your new response and updated score**
> >
> > Dear Reviewer,
> >
> > Thank you so much for your careful and insightful feedback and for raising the score! We are glad that our responses addressed your concerns. Your advice is inspiring and has provided us with guidance towards a more comprehensive paper. We truly appreciate your support for our work.

---

### Author Response · Authors · 2024-08-22
**Rebuttal Summary**

We thank all reviewers for their dedication to our paper and insightful comments, and we believe these comments are significant for improving the overall quality of our paper. We are encouraged that the reviewers appreciated our paper from various aspects, including its interesting findings (Reviewer 7nBB), comprehensive evaluations (Reviewer 7nBB, nFJQ, kTbb, nhq4), solid experiments (Reviewer 7nBB, nFJQ, kTbb), clear writing (Reviewer 7nBB, kTbb, nhq4), and reproducibility (Reviewer kTbb).

We have revised our paper according to the reviewers' well-taken comments. We would like to summarize our responses to some common concerns raised by various reviewers as follows:

- **Clarify the motivation for new task setting and knowledge source**. We highlight that our work focuses on **concept-level unlearning**, which is distinct from traditional instance-level unlearning. Concept-level unlearning is significantly more challenging as it requires the removal of all knowledge associated with a particular concept from the model, rather than deleting single data points. For the **task setting**, we do not provide a dedicated forget corpus because knowledge about a particular concept can be derived from multiple training data points, making it nearly impossible to identify all relevant data, much like searching for a needle in a haystack. For the **knowledge source**, we believe concept-level unlearning is a worthwhile problem to investigate, as there are situations where it's necessary to remove all knowledge related to a harmful or biased concept, a specific historical event, or a sensitive entity to ensure compliance with ethical standards or privacy regulations. (Reviewer nFJQ, 7nBB)

- **Assess the quality of the generated probes**. We evaluate the probes from the perspectives of **diversity** and **correctness**. The experimental results demonstrate that the probes we constructed are of high quality. Besides, we classify the knowledge points in the probes and find these knowledge points are meaningful entities rather than simple nouns or terms. (Reviewer nFJQ, 7nBB)

- **Increase the diversity of knowledge sources**. We construct **RWKU-Event** dataset containing 50 historical events as unlearning targets, including socially detrimental events such as terrorist attacks, massacres, murders, and political scandals. We have conducted experiments on this dataset, and the results aligned with those from the famous people dataset, further confirming the generalizability of our findings. (Reviewer kTbb, nhq4)

- **Analyze the impact of unlearning on long-tail entities**. We select 20 mid-popularity entities as unlearning targets and create the **RWKU-Tail** dataset using the same workflow. We observe that unlearning long-tail knowledge is easier and has a smaller impact on neighbor knowledge. This highlights that unlearning popular knowledge in RWKU is more challenging, requiring the development of more effective unlearning methods to minimize side effects. (Reviewer 7nBB)

We have also made revisions sincerely to address all the reviewers’ concerns, and provided detailed answers to each of your comments below. Given the enhancements based on the feedback, we believe our paper can become a more comprehensive and competitive work! Your insights are highly valuable to us, and we would greatly appreciate your feedback if you have any remaining concerns or suggestions. Please feel free to share with us, as we are more than willing to discuss with you and provide any additional clarifications as needed.

---

### Comment · Area_Chair_mJ4n · 2024-08-31
**Please address the author rebuttals**

Dear reviewer nFJQ, nhq4,

Thank you for your hard work.

As the discussion period will end soon (within 12 hours), I strongly encourage reviewers to acknowledge the rebuttals made by the authors.

It seems many reviews had common concerns and questions (which is good) and the author's responses convinced some of the reviewers; it would be worthwhile to go over the response to see whether it clarifies your concerns as well.

Let's make this last effort to make the reviewing process more productive!

Best,

---

### Decision · Program_Chairs · 2024-09-26

**Decision:**

Accept (Poster)

**Comment:**

The paper introduces the RWKU benchmark to measure the ability of large language models (LLMs) to unlearn specific knowledge. It focuses on removing knowledge about 200 famous people without access to a "forget" or "retain" corpus and evaluates model utility post-unlearning. The benchmark uses four Membership Inference Attack (MIA) methods and nine adversarial probes to measure how well models unlearn this knowledge.

The reviewers mostly agree that this paper touches upon critical issue of privacy and regulatory concerns with clear writing and that RWKU is a comprehensive benchmark.

There were concerns about the quality assessments of generated probes and requests for motivation for the new task setting and knowledge source. The authors have successfully convinced authors with some new analyses and clarificaitons.
Please include presented materials in the rebuttal and missing related works from the reviewers in your camera ready.